# Simple Degree-of-Freedom Modeling of the Random Fluctuation Arising in Human–Bicycle Balance

**Katsutoshi Yoshida \***  , **Keishi Sato and Yoshikazu Yamanaka**

Department of Mechanical and Intelligent Engineering, Utsunomiya University, Tochigi 321-8585, Japan;
sa-to@katzlab.jp (K.S.); yyamanaka@cc.utsunomiya-u.ac.jp (Y.Y.)
**\*** Correspondence: yoshidak@cc.utsunomiya-u.ac.jp

**Abstract:** In this study, we propose a new simple degree-of-freedom fluctuation model that accurately reproduces the probability density functions (PDFs) of human–bicycle balance motions as simply as possible. First, we measure the time series of the roll angular displacement and velocity of human–bicycle balance motions and construct their PDFs. Next, using these PDFs as training data, we identify the model parameters by means of particle swarm optimization; in particular, we minimize the Kolmogorov–Smirnov distance between the human PDFs from the participants and the PDFs simulated by our model. The resulting PDF fitnesses were over 98.7% for all participants, indicating that our simulated PDFs were in close agreement with human PDFs. Furthermore, the Kolmogorov–Smirnov statistical hypothesis testing was applied to the resulting human–bicycle fluctuation model, showing that the measured time responses were much better supported by our model than the Gaussian distribution.

**Keywords:** human–bicycle balance; stochastic model; identification; probability density function; particle swarm optimization

## 1. Introduction

Bicycles provide a useful means of short-distance transportation, and their utilization is expected to contribute to building a healthy and environmentally friendly society [1]. However, at least on Japanese roads, bicycle transportation is not always necessarily safe due to collisions with automobiles. The Japan National Police Agency reported that over 83% of bicycle accidents in Japan in the last ten years have involved automobiles [2]. To avoid such accidents, autonomous vehicle technology will play an important role; if it can predict bicycle motions, the resulting self-driving cars may reduce such accidents. For this purpose, accurate simulation models of bicycle motions are required, and they should be provided as simply as possible for the potential use of electronic control units in self-driving cars.

Bicycle motions with human riders in traffic seem to be broadly classified into two types: voluntary and involuntary. The former comprises purposeful motions such as right and left turning at a street intersection. Google [3] reported that their sensors can detect a cyclists' hand signals as an indication of an intention to make a turn or shift over. The latter comprises unconscious motions such as human fluctuated balance motions, which have already been found universally in human quiet standing [4–6], human stick balancing [7,8], human visuomotor tracking [9–11], and so on.

In this study, we propose a simple stochastic model that allows us to simulate the latter type of bicycle motion, i.e., involuntary fluctuated human–bicycle balance motions. To this end, we have conducted an experiment in which each human participant rides a bicycle on bicycle-trainer rollers, allowing it to move without rolling or yawing constraints. During this experiment, we measured the bicycle's rolling motion using a three-dimensional motion sensor attached to it. The measured

time-series of the rolling motion are characterized by their probability density functions (PDFs). Next, we designed our proposed human–bicycle model as a simple pendulum mechanism controlled by our human controller model, which was successfully used in our previous study [11] to simulate random human fluctuations during a visuomotor tracking task. We then identified the model parameters based on the measured PDFs as training data, using particle swarm optimization (PSO) to minimize the Kolmogorov–Smirnov (KS) distance between the measured PDFs and those simulated by our proposed human–bicycle model. The results show that our proposed model successfully reproduces the measured PDFs with fitnesses of over 98.7%. Furthermore, we conducted a statistical hypothesis test called the KS test [12,13] on our results to check their stochastic reliability, showing that the measured time series were much better supported by our model than the Gaussian distribution.

Our stochastic human-modeling approach sharply contrasts with other studies in the fields of autonomous or unmanned bicycle-control systems [14–22] because their models have been deterministic and not designed to have randomly fluctuating terms. Our approach also differs from Google's study on voluntary bicycle motions, as mentioned above [3]. Although there has been another study on the stochastic modeling of bicycle fluctuated motion [23], it addressed large-scale bicycle running paths, unlike our study, which deal with small fluctuations.

The rest of the paper is structured as follows: Section 2 describes the experimental test of human–bicycle balance motions. In Section 3, our proposed stochastic model of these motions is presented. In Sections 4 and 5, we describe the method of parameter identification and the identification results are presented with the KS testing results. Section 6 concludes our study.

## 2. Human–Bicycle Balance Experiment

### 2.1. Experimental Setup and Procedure

Figure 1 shows a photograph of our experimental device, a participant, and an experimenter. The experimental device consists of four units: a bicycle (BE-ELL03, Panasonic, Osaka, Japan), a set of bicycle-trainer rollers (E-MOTION, Elite, Italy), a three-dimensional motion sensor (CSM-MG100, Tokyo Aircraft Instrument, Tokyo, Japan), and a computer.

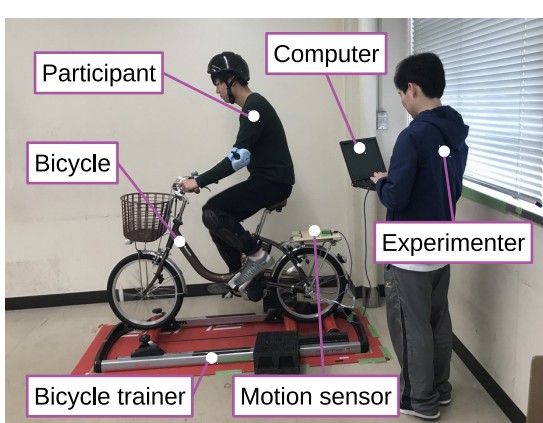

**Figure 1.** Photograph of our experimental device, a human participant, and an experimenter.

The experimental participants were eight healthy males in their early twenties. They were first instructed on the operation of the experimental device, the number of trials, and the duration of each trial. The experiment was performed according to the principles of the Declaration of Helsinki, and informed consent was obtained.

In each trial, the participant rode a bicycle on trainer rollers for significantly more than 180 s at a speed of about 15 km/h. Several practice trials were performed prior to measurement. The air pressure of the tires was set to 300 kPa.

Note that, in our setup, the bicycle is constrained on the stationary trainer. This may cause our measured motion to differ from free motions on the road. Such potential discrepancy, however, did permit a three-minute safety trial without external disturbances and was minimized by using a roller-type trainer known to have fewer constraints than other types of trainers.

## 2.2. Experimental Data

During each trial, the time series of the measurement vector,

$$\boldsymbol{x}(t) = (x_1(t), x_2(t))^T := (\theta(t), \dot{\theta}(t))^T \tag{1}$$

(hereafter, $(\cdot)^T$ denotes a transpose), was obtained by the motion sensor. Here, $\theta$ [rad] is the roll angle from the vertical line to the bicycle's vertical axis, and $\dot{\theta} := d\theta/dt$ [rad/s] is the corresponding angular velocity. Figure 2 schematically shows the definition of the roll angle in the front view of the bicycle. The bicycle's vertical axis was nominally determined based on the direction of the bicycle frame. The motion sensor was initially calibrated to output $\theta = 0$ when the bicycle's vertical axis was parallel to the vertical line. Therefore, $\theta = 0$ does not indicate the upright equilibrium of the unmanned bicycle. Nevertheless, since the resulting calibration setting was commonly maintained for all participants and trials, the obtained datasets can be compared with each other.

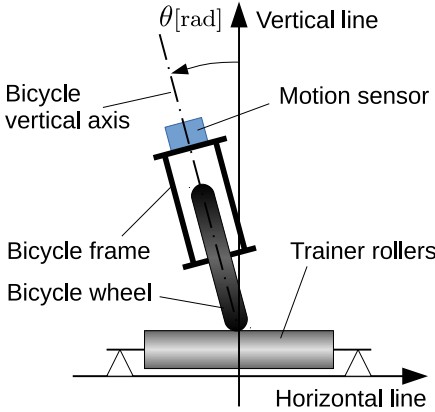

**Figure 2.** Schematic front view of the bicycle during the experiment.

Throughout the experiment, $\boldsymbol{x}(t)$ were stored in the computer in the following form:

$$\{\boldsymbol{x}_{hum}^{(s,n)}(t_0), \cdots, \boldsymbol{x}_{hum}^{(s,n)}(t_i), \cdots, \boldsymbol{x}_{hum}^{(s,n)}(t_{I-1})\}, \quad i = 0, \cdots, I-1, \ s = 1, \cdots, S, \ n = 1, \cdots, N, \tag{2}$$

where $t_i := i\Delta t$ [s] is a discrete time with a sampling period of $\Delta t$ [s], $I$ is the length of the time series, $s$ and $S$ are an index and the number of participants, respectively, and $n$ and $N$ are an index and the number of trials, respectively.

For this study, we chose $\Delta t = 10^{-2}$ s and $I = 18{,}001$ to obtain the physical data length $(I-1)\Delta t = 180$ s. The number of participants was $S = 8$, and the number of trials undertaken by each participant was $N = 5$.

Figure 3 shows the measured time series for $(s, n) = (1, 1)$, i.e., for the first participant's first trial. The result clearly exhibits fluctuations specific to human balancing motions in which large-amplitude spikes intermittently arise among the moderate-amplitude fluctuation process, which has already been recognized as temporal intermittency in the field of nonlinear physics [6–8]. Thus, we obtained the time series for all $s = 1, \cdots, S$ and $n = 1, \cdots, N$.

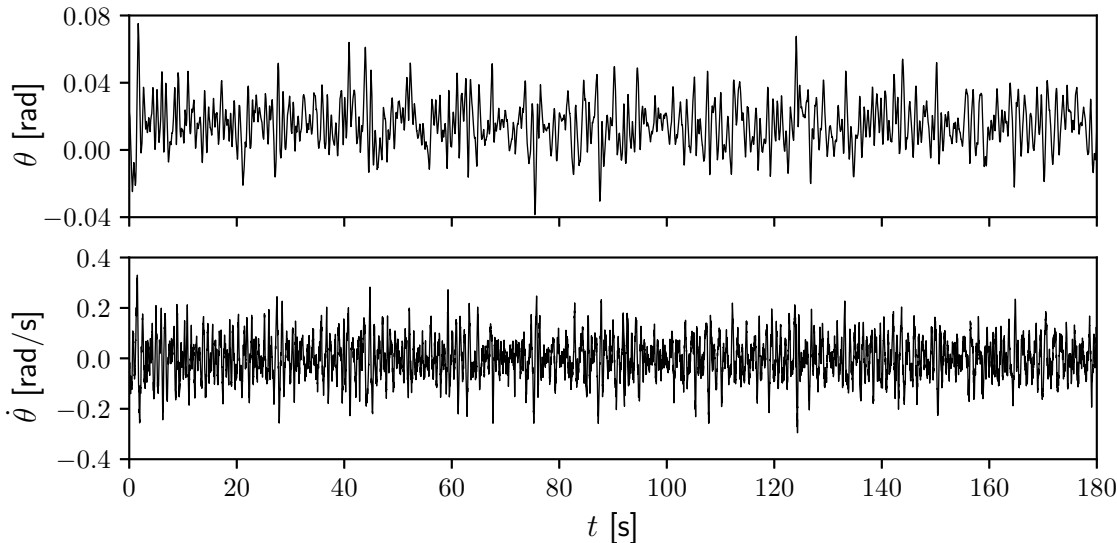

**Figure 3.** The measured time series of the human–bicycle balance for $(s, n) = (1, 1)$.

As shown in the upper graph of Figure 3, the vibrational center of $\theta(t)$ is shifted from the origin, $\theta = 0$. This is mainly because the calibrated $\theta = 0$ does not indicate the vertical equilibrium of the bicycle, as mentioned above. We statistically evaluate these shifts by identifying them as the mean roll angles given by

$$E^{(s)}[\theta] := \frac{1}{N \times I} \sum_{n=1}^{N} \sum_{i=0}^{I-1} x_1^{(s,n)}(t_i), \quad s = 1, \cdots, S, \tag{3}$$

which is the temporal average of the $s$th participant's $\theta(t)$, further averaged over all of his trials. The resulting values are listed in Table 1 with standard deviations given by

$$SD^{(s)}[\theta] = \sqrt{V^{(s)}[\theta]}, \quad V^{(s)}[\theta] := \frac{1}{N \times I + 1} \sum_{n=1}^{N} \sum_{i=0}^{I-1} \left( x_1^{(s,n)}(t_i) - E^{(s)}[\theta] \right)^2, \quad s = 1, \cdots, S. \tag{4}$$

Viewing the average over all participants' values, $E^{(s)}[\theta]$ takes about $1.75 \times 10^{-2}$, indicating that the actual equilibrium angle of the bicycle was approximately at $\theta \approx 1.75 \times 10^{-2}$ rad (or $1.01°$) in our setup. In addition, each $E^{(s)}[\theta]$ value is slightly different due to the respective riding forms of the participants.

**Table 1.** The mean roll angle of the $s$th participant.

| $s$ | 1 | 2 | 3 | 4 |
|---|---|---|---|---|
| $E^{(s)}[\theta]$ | $1.71 \times 10^{-2}$ | $1.55 \times 10^{-2}$ | $1.71 \times 10^{-2}$ | $1.69 \times 10^{-2}$ |
| $SD^{(s)}[\theta]$ | $1.46 \times 10^{-2}$ | $1.33 \times 10^{-2}$ | $1.63 \times 10^{-2}$ | $1.07 \times 10^{-2}$ |
| $s$ | 5 | 6 | 7 | 8 |
| $E^{(s)}[\theta]$ | $1.59 \times 10^{-2}$ | $1.65 \times 10^{-2}$ | $1.83 \times 10^{-2}$ | $2.31 \times 10^{-2}$ |
| $SD^{(s)}[\theta]$ | $1.40 \times 10^{-2}$ | $1.37 \times 10^{-2}$ | $1.30 \times 10^{-2}$ | $1.42 \times 10^{-2}$ |

*2.3. Construction of Measured PDFs*

First, we obtain $P_{hum}^{(s,n)}(x_1, x_2)$, the joint PDF with respect to the components of the time series in (2) for the $s$th participant's $n$th trial, by normalizing the two-dimensional histogram of $\{x^{(s,n)}(t_i)\}_{i=0}^{I-1}$ with bin width $(\overline{x_k} - \underline{x_k}) / N_{\text{bin}}$ $(k = 1, 2)$. Here, $N_{\text{bin}}$ is the number of histogram bins and $\overline{x_k}$ and $\underline{x_k}$ are the upper and lower limits of $x_k$, respectively.

Next, the resulting $P_{hum}^{(s,n)}(x_1, x_2)$ is averaged over all trials $n = 1, \cdots, N$ to obtain the $s$th participant's joint PDF as

$$P_{hum}^{(s)}(x_1, x_2) = \frac{1}{N} \sum_{n=1}^{N} P_{hum}^{(s,n)}(x_1, x_2). \tag{5}$$

We call (5) the measured PDF of the $s$th participant.

Figure 4 shows the measured PDF of all participants ($s = 1, \cdots, 8$). In this study, we commonly set $N_{\text{bin}} = 40$, $(\underline{x_1}, \overline{x_1}) = (-0.04, 0.08)$, and $(\underline{x_2}, \overline{x_2}) = (-0.3, 0.3)$ for all PDFs , as was done in Figure 4.

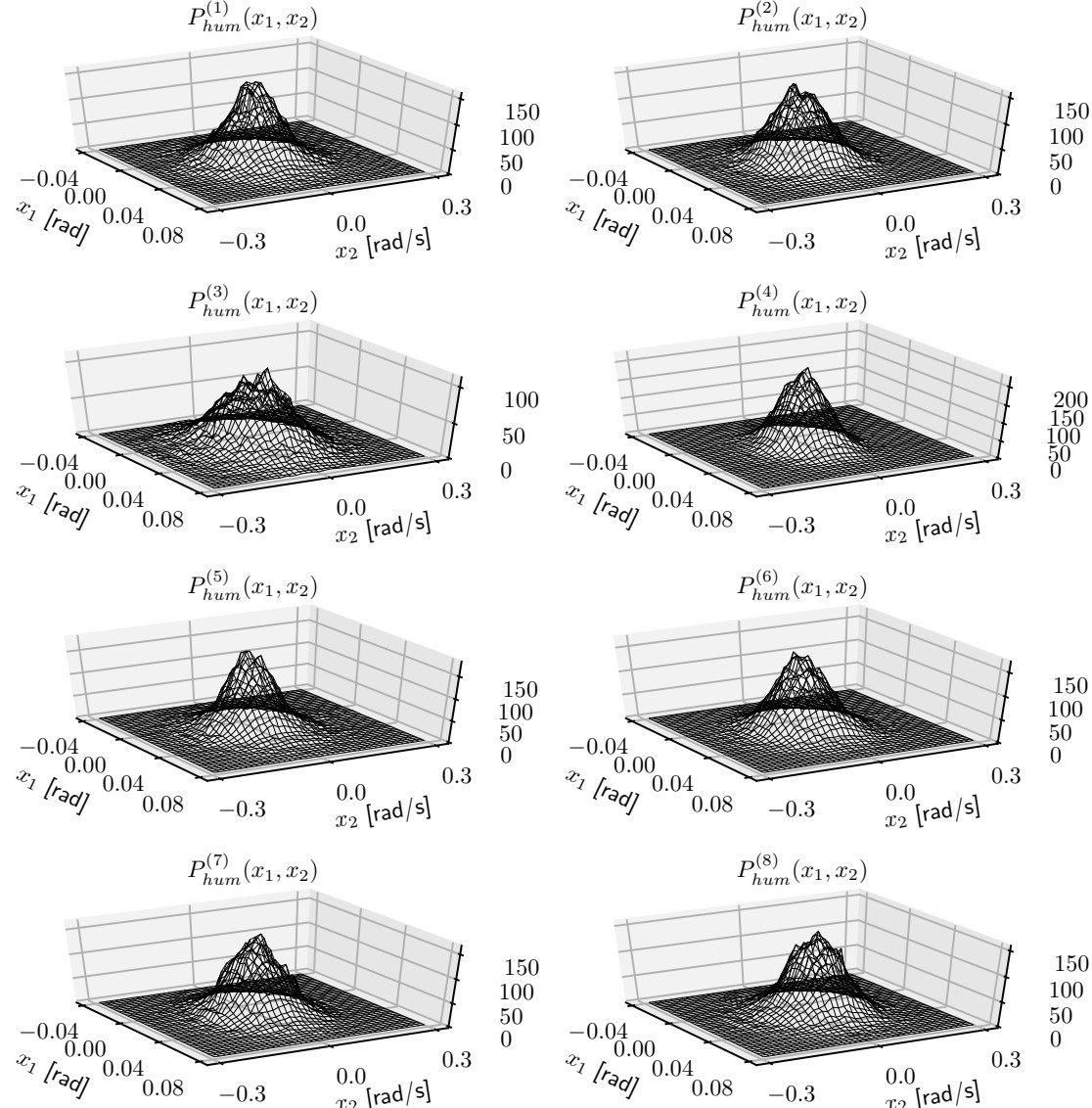

**Figure 4.** The measured joint probability density functions (PDFs) from all participants ($s = 1, \cdots, 8$).

## 3. Fluctuation Model of the Human–Bicycle Balance

In this section, we propose a new fluctuation model that allows us to reproduce the measured PDFs obtained above as simply as possible.

### 3.1. A Human–Bicycle Fluctuation Model

In view of our setup in Figure 2, we model the human–bicycle mechanics by a simple inverted pendulum about the contact point of the bicycle wheel. Based on the time series statistics in Table 1,

we also assume that the roll angle $\theta$ about the equilibrium is sufficiently small; the maximal three standard deviation indicates $3 \times \max_s SD^{(s)}[\theta] \approx 4.88 \times 10^{-2}$ rad (or $2.80°$ deg). Therefore, we model the bicycle's rolling motion by a linearized inverted pendulum of the form:

$$mr^2\ddot{\theta} - mgr(\theta - \hat{\theta}) = T, \tag{6}$$

where $m$ [kg] and $r$ [m] are the mass and length of the pendulum, respectively, $g := 9.81$ m/s$^2$ is the gravitational acceleration, $T$ [Nm] is a torque input, and $\hat{\theta}$ [rad] is the equilibrium angle. The equation of motion (6) is nondimensionalized and represented in the following state-space form:

$$\dot{x}_1 = x_2, \quad \dot{x}_2 = k(x_1 - \hat{\theta}) + u, \tag{7}$$

where $\boldsymbol{x} = (x_1, x_2)^T := (\theta, \dot{\theta})^T$, $k := g/r$, and $u := T/mr^2$.

Next, we specify $u$ to simulate the human fluctuation during human–bicycle motion. As successfully demonstrated in our previous study [11], some human fluctuations can be accurately simulated by the following state-feedback mechanism:

$$u = F_1\{1 + \sigma_1\xi_1(t)\}(x_1 - \hat{\theta}) + F_2 x_2 + \sigma_2\xi_2(t), \tag{8}$$

where $\xi_1(t)$ and $\xi_2(t)$ are independent white Gaussian noises with zero mean and unit variance. The first coefficient $F_1\{1 + \sigma_1\xi_1(t)\}$ represents a random proportional gain with mean $F_1$ and variance $(F_1\sigma_1)^2$. The second coefficient $F_2$ is a deterministic derivative gain. The third term $\sigma_2\xi_2(t)$ represents an additive random perturbation.

Finally, we substitute (8) into (7), rewrite it to reduce the dependency between parameters, and propose a human–bicycle fluctuation model of the form

$$\dot{x}_1 = x_2, \quad \dot{x}_2 = \{p_1 + p_2\xi_1(t)\}(x_1 - \hat{\theta}) + p_3 x_2 + p_4\xi_2(t), \tag{9}$$

with the generalized parameter vector

$$\boldsymbol{p} = (p_1, p_2, p_3, p_4) := (g/r + F_1, F_1\sigma_1, F_2, \sigma_2) \in \mathbb{R}^4, \tag{10}$$

which parameterizes the equivalent properties of human–bicycle motion. Here, $p_1$ and $p_2$ are a mean and a standard deviation of the randomly fluctuating stiffness of the human–bicycle motion, respectively, $p_3$ is a deterministic viscous damping, and $p_4$ is an additive random fluctuation strength.

*3.2. Calculation of Simulated PDFs*

Using given $\hat{\theta}$ and $\boldsymbol{p}$, we obtain $N'$ samples of the stationary numerical solution of (9) as

$$\{\boldsymbol{x}_A^{(n')}(t_i; \hat{\theta}, \boldsymbol{p})\}_{i=0}^{I-1}, \quad n' = 1, \cdots, N', \tag{11}$$

by means of a fourth-order Runge–Kutta–Gill method with time step $\Delta t = 10^{-2}$ s, the same as the experimental sampling period. To generate these samples, $N'$ different sequences of normal pseudo-random numbers [24],

$$\{v_i^{(n')}\}_{i=0}^{I-1}, \quad n' = 1, \cdots, N', \tag{12}$$

are used to simulate the independent white Gaussian noises $\xi_1(t)$ and $\xi_2(t)$ by

$$\xi_l(t_i) \approx v_i^{(n')}(\Delta t)^{-1/2}, \quad l = 1, 2, \tag{13}$$

where $(\Delta t)^{-1/2}$ is the numerical factor required for integrating stochastic differential equations [25].

From the simulated time series in (11), we construct the $n'$th sample's joint PDF $P_{sim}^{(n')}(x_1, x_2; \hat{\theta}, \boldsymbol{p})$ using the same procedure and conditions as applied for deriving the measured PDF in Section 2.3. We also take the average $P_{sim}^{(n')}(x_1, x_2; \hat{\theta}, \boldsymbol{p})$ over all samples by

$$P_{sim}(x_1, x_2; \hat{\theta}, \boldsymbol{p}) = \frac{1}{N'} \sum_{n'=1}^{N'} P_{sim}^{(n')}(x_1, x_2; \hat{\theta}, \boldsymbol{p}). \tag{14}$$

We call (14) the simulated PDFs for $\hat{\theta}$ and $\boldsymbol{p}$, which are to be compared with the measured $P_{hum}^{(s)}(x_1, x_2)$.

## 4. Method of Parameter Identification

In this section, we formulate the identification problem of an unknown parameter vector $\boldsymbol{p}$ that allows the simulated $P_{sim}(x_1, x_2; \hat{\theta}, \boldsymbol{p})$ to reproduce the measured $P_{hum}^{(s)}(x_1, x_2)$.

### 4.1. Parameter Identification Problem

We solve the optimization problem

$$\underset{\boldsymbol{p}}{\text{Minimize}}\, E(\boldsymbol{p}) \tag{15}$$

with the cost function

$$E(\boldsymbol{p}) := \max_{(x_1, x_2, i)} \left\{ |F_{sim}^i(x_1, x_2; \hat{\theta}, \boldsymbol{p}) - F_{hum}^{i(s)}(x_1, x_2)| \right\}. \tag{16}$$

Equation (16) is known as a two-dimensional Kolmogorov–Smirnov (KS) distance [12] and is used for two-dimensional goodness-of-fit testing between the empirical distribution of data and a hypothetical density law or between two distributions of separate data [12,13]. Here, $F_{sim}^i(x_1, x_2)$ and $F_{hum}^{i(s)}(x_1, x_2)$ ($i = 1, \cdots, 4$) are the cumulative distribution functions (CDFs) with respect to the four quadrants about $(x_1, x_2)$ on the $(x_1, x_2)$-plane, i.e.,

$$F_{sim}^i(x_1, x_2; \hat{\theta}, \boldsymbol{p}) := \iint_{R_i} P_{sim}(x_1, x_2; \hat{\theta}, \boldsymbol{p}) dx_1 dx_2, \quad F_{hum}^{i(s)}(x_1, x_2; \hat{\theta}, \boldsymbol{p}) := \iint_{R_i} P_{hum}^{(s)}(x_1, x_2) dx_1 dx_2, \tag{17}$$

with their domains $R_1 := [\underline{x_1}, x_1] \times [\underline{x_2}, x_2]$, $R_2 := [x_1, \overline{x_1}] \times [\underline{x_2}, x_2]$, $R_3 := [\underline{x_1}, x_1] \times [x_2, \overline{x_2}]$, and $R_4 := [x_1, \overline{x_1}] \times [x_2, \overline{x_2}]$, respectively. The cost function (16) evaluates the CDFs' reproduction error and satisfies $E(\boldsymbol{p}) = 0$ if $P_{sim}(x_1, x_2; \hat{\theta}, \boldsymbol{p}) = P_{hum}^{(s)}(x_1, x_2)$. Hence, it also indicates the PDFs' reproduction error.

### 4.2. Particle Swarm Optimization (PSO)

We employ PSO [26] to solve (15). Consider a swarm of $M$ candidate solutions,

$$\{\boldsymbol{p}^1, \boldsymbol{p}^2, \cdots, \boldsymbol{p}^i, \cdots, \boldsymbol{p}^M\}, \quad \boldsymbol{p}^i \in \mathbb{R}^4, \tag{18}$$

which are called particles. Each component of $\boldsymbol{p}^i$ is recursively updated by

$$\begin{cases} v_j^i(k+1) = \omega v_j^i(k) + c_1 \eta_{1j}^i(k) \left( pb_j^i(k) - p_j^i(k) \right) + c_2 \eta_{2j}^i(k) \left( gb_j(k) - p_j^i(k) \right), \\ p_j^i(k+1) = p_j^i(k) + v_j^i(k+1), \quad (k = 0, 1, \cdots, K), \end{cases} \tag{19}$$

where $p_j^i(k)$ denotes the $j$th component of $\boldsymbol{p}_i$ at iteration $k$; $v_j^i(k)$ is the corresponding velocity; $\omega$, $c_1$ and $c_2$ are system parameters of PSO; $\eta_{1j}^i(k)$ and $\eta_{2j}^i(k)$ are random numbers independently generated

by [24] for each $i$, $j$, and $k$ with a uniform distribution in $[0, 1]$; and $pb_j^i(k)$ and $gb_j(k)$ are the $j$th components of the vectors $\boldsymbol{pb}_i(k)$ and $\boldsymbol{gb}(k) \in \mathbb{R}^4$, respectively.

$\boldsymbol{pb}_i(k)$ is the position of the particle taking the lowest cost among those at $\boldsymbol{p}_i(0), \cdots, \boldsymbol{p}_i(k)$; this is called the personal best. $\boldsymbol{gb}(k)$ is the position of the particle with the lowest cost among all particles for all iterations up to $k$; this is called the global best. For sufficiently large $K$, $\boldsymbol{gb}(K)$ is expected to be close to the optimal solution $\boldsymbol{p}^*$.

## 5. Identification Results

### 5.1. Identification Condition

In our PSO application, the number of particles was set to $M = 32$, and the initial particles $\boldsymbol{p}^i(0)$, $i = 1, \cdots, M$ were given by random points uniformly distributed within the four-dimensional hyperrectangle

$$\boldsymbol{D}_0 := \{(p_1, p_2, p_3, p_4) \mid -30 \le p_1 \le 0,\ 0 \le p_2 \le 3,\ -10 \le p_3 \le 0,\ 0 \le p_4 \le 3\}. \tag{20}$$

The number of iterations is $K = 500$. We set the number of the simulated samples in (14) to $N' = 5$, the same as the number of experimental trials par participant.

We hereafter denote by $\boldsymbol{p}^{(s)}$ the optimized solution $\boldsymbol{gb}(K)$ obtained from the $s$th participant's data: $P_{hum}^{(s)}(x_1, x_2)$ and $\hat{\theta} = \hat{\theta}^{(s)} := E^{(s)}[\theta]$. We also use the notation $P_{sim}^{(s)}(x_1, x_2) := P_{sim}(x_1, x_2; \hat{\theta}^{(s)}, \boldsymbol{p}^{(s)})$ for the simulated PDFs derived from the $s$th participant's data.

### 5.2. Identification Results

Table 2 lists the identified vector components of $\boldsymbol{p}^{(s)}$ by PSO for all participants $s = 1, \cdots, 8$ and the corresponding KS cost value $E(\boldsymbol{p}^{(s)})$. The seventh column indicates the cost value as a PDF fitness value of the form

$$\text{Fitness} := (1 - E) \times 100\,\%, \tag{21}$$

which indicates the accuracy of our human–bicycle fluctuation model (9) in terms of the reproducibility of PDFs. The best and worst results are indicated by "**" and "*", respectively. The second-last and the last columns show the cost value and the corresponding fitness value, respectively, between the measured $P_{hum}^{(s)}(x_1, x_2)$ and the mathematical two-dimensional Gaussian PDF $P_{Gauss}^{(s)}(x_1, x_2)$ with the same mean vector and covariance matrix as those of the measured $P_{hum}^{(s)}(x_1, x_2)$.

**Table 2.** Identified $\boldsymbol{p}^{(s)} = (p_1, p_2, p_3, p_4)$ and its cost value for the $s$th participant. "**" denotes the best result and "*", the worst. The last two columns show the corresponding Gaussian results.

| | Our Proposed Fitting | | | | | | Gaussian Fitting | |
|---|---|---|---|---|---|---|---|---|
| $s$ | $p_1$ | $p_2$ | $p_3$ | $p_4$ | $E(\boldsymbol{p}^{(s)})$ | Fitness | $E$ | Fitness |
| 1 | $-32.6272$ | $3.35867$ | $-5.84823$ | $2.74614$ | $9.611 \times 10^{-3}$ | 99.04% | $1.134 \times 10^{-1}$ | 88.66% |
| 2 | $-39.6902$ | $1.21198$ | $-1.72440$ | $1.53507$ | $7.189 \times 10^{-3}$ | 99.28% | $1.684 \times 10^{-1}$ | 83.16% |
| 3 | $-44.5088$ | $1.08744$ | $-5.97971$ | $3.70913$ | $7.056 \times 10^{-3}$ | 99.29% | $1.234 \times 10^{-1}$ | 87.66% |
| 4 | $-46.6330$ | $1.17335$ | $-6.31916$ | $2.58083$ | $7.478 \times 10^{-3}$ | 99.25% | $1.652 \times 10^{-1}$ | 83.48% |
| 5 | $-37.5965$ | $1.61197$ | $-1.77080$ | $1.54042$ | $1.228 \times 10^{-2}$ | 98.77% * | $1.640 \times 10^{-1}$ | 83.60% |
| 6 | $-34.6138$ | $0.993204$ | $-7.01156$ | $2.96107$ | $5.689 \times 10^{-3}$ | 99.43% ** | $1.351 \times 10^{-1}$ | 86.49% |
| 7 | $-38.5872$ | $1.21683$ | $-7.31069$ | $3.07624$ | $6.011 \times 10^{-3}$ | 99.40% | $9.134 \times 10^{-2}$ | 90.87% |
| 8 | $-32.5560$ | $2.55268$ | $-2.15329$ | $1.63554$ | $8.500 \times 10^{-3}$ | 99.15% | $1.225 \times 10^{-1}$ | 87.75% |

The results clearly show that our proposed model (9) successfully achieved over 98.7% PDF fitness, even in the worst case ($s = 5$). This implies that it provides much better fitness than conventional Gaussian models.

Next, the left column of Figure 5 shows the difference of our simulated $P_{sim}^{(s)}(x_1, x_2)$ from the measured $P_{hum}^{(s)}(x_1, x_2)$, and the right column shows that for the Gaussian PDF $P_{Gauss}^{(s)}(x_1, x_2)$, for all $s$. The plot ranges are the same as those of all plots.

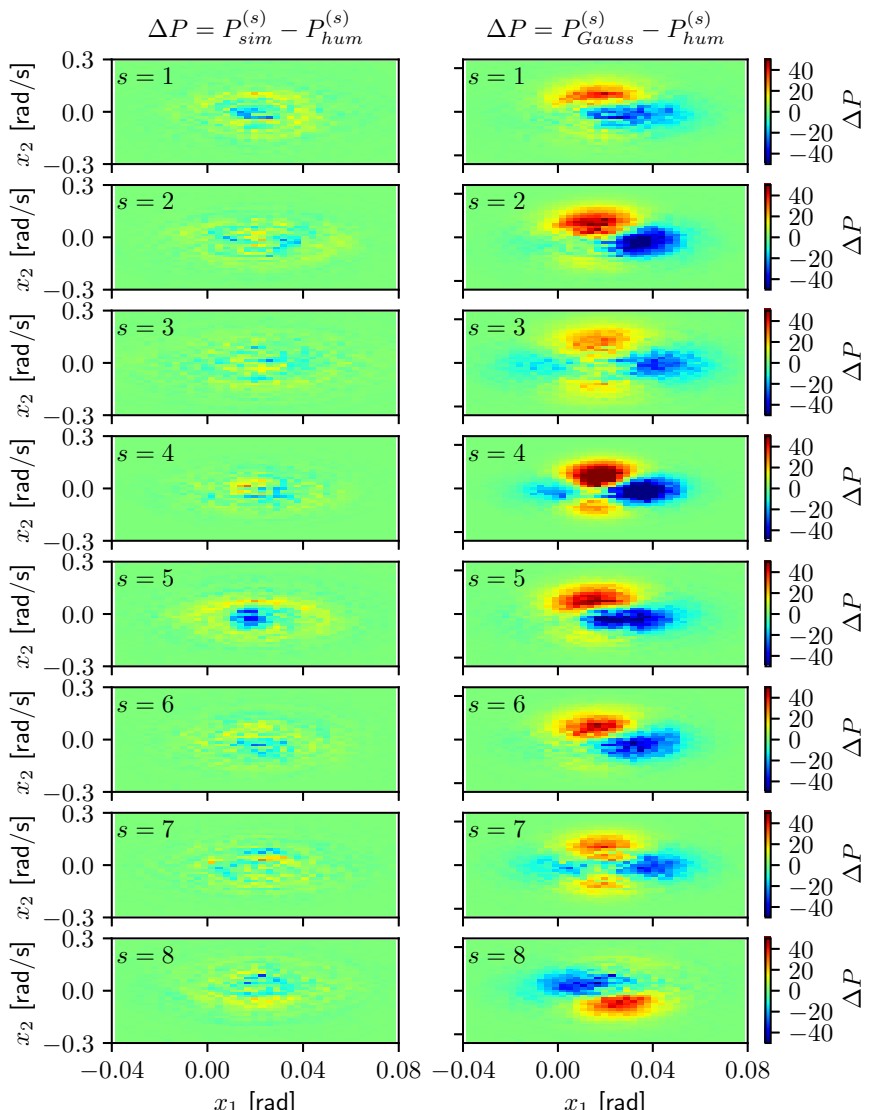

**Figure 5.** Difference of our simulated $P_{sim}^{(s)}(x_1, x_2)$ from the measured $P_{hum}^{(s)}(x_1, x_2)$ (left column) and that of the equivalent Gaussian PDF $P_{Gauss}^{(s)}(x_1, x_2)$ (right column), for all $s$.

As already indicated by the fitness values in Table 2, our simulated PDFs agree well with the measured PDFs, compared to the Gaussian PDFs. As the shapes of Gaussian PDFs are symmetric by definition, the right column's results imply that our measured PDFs have asymmetric shapes, as indicated by the deep-colored peaks and troughs.

Therefore, our proposed model properly reproduced such asymmetric shapes, which were barely reproduced by the Gaussian distribution.

*5.3. KS Test*

Up to this point, we have found that our simple model (9) provides high reproducibility of the PDF shapes of the human–bicycle fluctuation. However, this does not directly support the stochastic

reliability of our model. Therefore, in this section, we conduct a statistical-hypothesis test called the KS test [12,13] on our results to check their stochastic reliability.

As described in [12], the statistic of the one-sample KS test is provided by

$$Z(n, E) = n^{1/2}E, \tag{22}$$

where $E$ is the KS distance already given in (16), and $n$ is the number of samples. On the other hand, the two-sample KS test employs the following statistic:

$$Z(n_1, n_2, E) = \left(\frac{n_1 n_2}{n_1 + n_2}\right)^{1/2} E, \tag{23}$$

where $n_1$ and $n_2$ are the numbers of two independent samples. Under the null hypothesis (the sample follows a given distribution or the two samples follow the same distribution) and for large $n$ (or $n_1$ and $n_2$), the random variable $Z$ follows the following CDF [12,13]:

$$F(z) := \text{Prob}(Z > z) = 2 \sum_{i=1}^{\infty} (-1)^{i-1} \exp(-2i^2 z^2), \tag{24}$$

where the lowercase $z$ denotes a value of $Z$. The value of $F(z)$ is called the *p*-value of the observed $z$. If this *p*-value is smaller than a pre-defined $\alpha$ called a significance level, the null hypothesis is rejected.

In Figure 6, the solid curve shows $F(z)$. The small circles plot the *p*-values between $P_{hum}^{(s)}$ and $P_{sim}^{(s)}$ at $z = Z(n_1, n_2, E)$ using $E = E(\mathbf{p}^{(s)})$ listed in Table 2, the measured data length $n_1 = I \times N = 18{,}001 \times 5$, and the simulated $n_2 = I \times N' = n_1$. Under the significance level $\alpha = 0.01$, the null hypothesis is rejected for $s = 1, 5,$ and $8$; that is to say that three of the eight measurements cannot be said to follow our simulated distributions. On the other hand, the cross marks show the results between $P_{hum}^{(s)}$ and $P_{Gauss}^{(s)}$ at $z = Z(n, E)$ using the $E$ values listed in Table 2 and $n = n_1$. In this Gaussian case, the null hypothesis is rejected for all $s$; i.e., no measurements can be said to follow these Gaussian distributions.

Given the above, we conclude in terms of statistical hypothesis testing that our simple model (9) can simulate the time series of human–bicycle fluctuations much better than the Gaussian distribution.

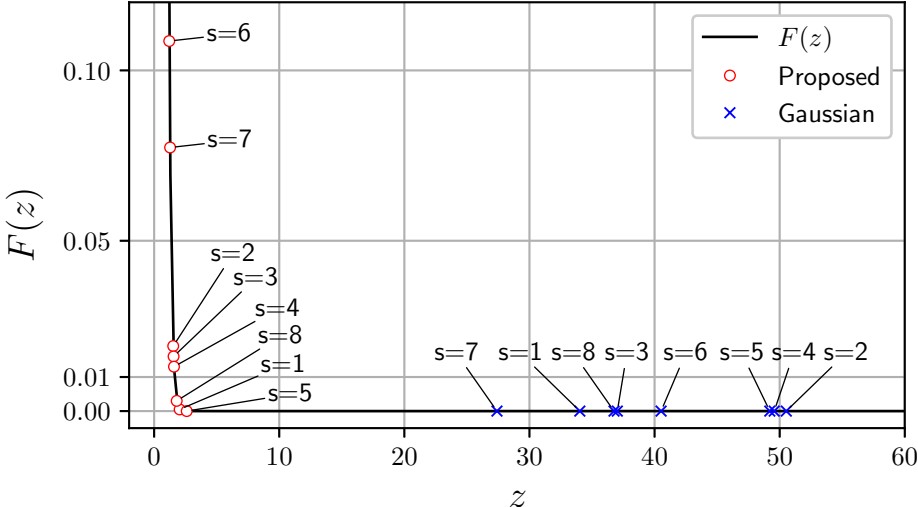

**Figure 6.** Kolmogorov–Smirnov (KS) testing results. The solid curve plots $F(z)$, the KS statistic cumulative distribution function (CDF). The small circles indicate the *p*-values between measured $P_{hum}^{(s)}$ and our proposed $P_{sim}^{(s)}$, and the cross marks indicate those between $P_{hum}^{(s)}$ and Gaussian $P_{Gauss}^{(s)}$.

Note that our proposed model can separately fit the measurements of individual participants; it does not provide a general description of human–bicycle fluctuation. Conversely, it provides a means of mechanical parameterization of individual difference. The obtained parameter vector $p$ is useful for comparing individuals and seeing how they differ, in a mechanical manner.

## 6. Conclusions

In this study, we have constructed a simple degree-of-freedom human–bicycle fluctuation model that accurately reproduces the PDFs of experimentally measured human–bicycle balance motions.

First, we measured the time series of the roll angular displacement and the velocity of the human–bicycle balance motions and constructed their PDFs. Using these PDFs as the training data, we identified the model parameters by PSO, minimizing the KS distance between the measured PDF from the participant and the simulated PDF from our model. The resulting PDF fitnesses were over 98.7%, indicating that the simulated PDFs were in close agreement with the measured ones.

Next, we applied the KS statistical hypothesis test to our results, showing that our model simulated the time series of human–bicycle fluctuation much better than the Gaussian distribution.

The above result leads to the conclusion that our proposed model can provide an accurate single-degree-of-freedom model of human–bicycle fluctuations.

In future work, using our model parameters, we plan to compare various cyclists of different ages and genders who ride different types of bicycles in different environments. We also plan to develop a multi-degree-of-freedom fluctuation model of human–bicycle balance motions, making it possible to simulate fluctuating bicycle running paths based on physically identified human–bicycle parameters.

**Author Contributions:** Conceptualization, K.Y.; methodology, K.Y. and Y.Y.; software, K.Y. and K.S.; validation, K.Y., K.S. and Y.Y.; formal analysis, K.Y. and Y.Y.; investigation, K.Y. and K.S.; resources, K.Y.; data curation, K.S.; writing—original draft preparation, K.Y.; writing—review and editing, K.Y.; visualization, K.Y. and K.S.; supervision, K.Y.; project administration, K.Y.; funding acquisition, K.Y. and Y.Y.

**Funding:** This work was funded by JSPS KAKENHI Grant Numbers JP18H01391 and JP17H06552.

**Acknowledgments:** We wish to express our gratitude to the members of the System Dynamics Group at Utsunomiya University for their participation and cooperation as participants in this study.

**Conflicts of Interest:** The funders had no role in the design of the study; in the collection, analyses, or interpretation of data; in the writing of the manuscript, or in the decision to publish the results.

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
