# Peer review of "Simple Degree-of-Freedom Modeling of the Random Fluctuation Arising in Human–Bicycle Balance"

_applsci, doi:10.3390/app9102154_

Round 1

Reviewer 1 Report

This paper proposes a simple one-degree-of freedom model for the balance motions of human cyclists. The paper essentially has two ingredients:

(1) an experimental study, which allows the measurements of the time series of the roll angle theta for human participants, measured on a bicycle trainer with motion sensors. The measured time series are used to produce empirical PDFs of the roll angle for each participant.

(2) a model of the time evolution of theta by means of an inverted pendulum with stochastic forcing. The model has four parameters (p1, p2, p3, p4). The model allows the construction of simulated PDFs by a forward solve with a 4th order Runge Kutta method. The parameters are tuned with particle swarm optimisation, the optimisation target is to bring the simulated PDFs as close as possible to the empirical PDFs. The authors report the parameter vectors for each participant and the PDF fitness, which is excellent (99.8 % worst case).

The paper is well written and the methodology is sound. The findings are novel and relevant. I recommend the paper for publication after minor review. Detailed comments below:

- the number of participants is N = 8, which is quite low. This raises questions about the generalisability of the findings, ideally the number of participants should be closer to 100 in order to get some statistical stability.

- it is difficult to judge for the reader how different the empirical PDFs for the different participants are, as the authors show only one example. It would be useful to provide a range of examples and a measure of similarity between the PDFs.

- in the experimental setup, the bicycle is constrained on a stationary trainer. It is thus questionable if the results will generalise to bicycle motion which is unconstrained (e.g. outdoor field experiment). The authors should comment on this.

- while the model very soundly reproduces the empirical PDFs, this does not (as the authors rightly point out in section 5.3) guarantee that the time responses of the dynamics are well approximated, especially so since the PDFs seem fairly well approximated by a normal distribution. The authors do show visualisations of both simulated and measured time series in section 5.3, but ultimately they do not provide a quantitative measure of how similar these time series are. This is not an easy problem due to the stochastic forcing, but the authors should elaborate here and at least give some further directions.

Author Response

Please refer to the PDF file.

Reviewer 2 Report

In the manuscript the authors conduct some experiments to analyse the fluctuations of a human-bicycle balance with interesting potential applications to cyclist behaviour prediction. The research and questions are interesting and the methods clearly described. Nevertheless I have some concerns about the analysis and conclusions of the study:

1) The authors use a 2 variable model to explain the fluctuations observed in the experimental data. Such model would entail a multivariate distribution P(x1,x2), nevertheless, if I understood correctly, the authors only use the marginal distributions P(x1) and P(x2) to fit the data. In order to claim that the physical model is accurate it would be necessary to reproduce the show that the whole distribution is recovered. 

2) In order to fit the data, the authors are comparing the empirical and the modelled distributions by binning the empirical data. This is not necessary and it may bias the results. The alternative is using the CDF and compare the distributions using a distance based on the CDF such as the Kolmogrov-Smirnov distance. 

3) The authors show that using a mechanistic model they can recover the observed data, nevertheless there is no proof that this is the correct model behind the data. For instance, there is no null hypothesis or alternative hypothesis to test the model proposed against other possibilities. For instance, one possibility would be to assume that the fluctuations are normally distributed. Showing that their model is able to reproduce the data better than other assumptions would give confidence on the model used. 

4) There are better statistical tests (Kolmogorov-Smirnov tests, Kuiper's test, ...) designed to check wether observed experimental data is compatible with an underlying distribution. This should be included in the analysis to state with quantitative statistical confidence the validity of the results. 

4) Following the same idea, the model is able to make some mechanical predictions. For instance it predicts how the different PDFs would change with the mass of the driver or the height of the bike. I understand that making such experiments now would be hard, but at least would be interesting to include such results from the fitted model to show which predictions does the model provide over a simple curve fitting. 

5) One of the aims of the authors is to create a model that is able to predict the details of a stochastic behaviour. Nevertheless, the authors show that they can fit the model to each individual. In order to have a predictive model, it is essential to have a comparison between individuals and see how they differ between them, or which are the errors when one tries to fit the model to all the individuals at once. 

6) The last section "reproducibility of time responses" is very vague. In order to test that the model is able to reproduce the time responses the authors should compare measurables of the autocorrelation of the different observables in time and, as before, compare it against other hypothesis such as white noise or an Ornstein-Ulenbeck process.

Other minor comments:

7) Some of the plots are missing the units. 

Author Response

Please refer to the PDF file.
